# Contrastive guidance and feedback: A Suitable way to improve 3D Consistency of Multiview Diffusion Model

## Abstract

Recently, diffusion models have shown potential in 3D generation tasks, and the novel view synthesis (NVS) task, a bridge between 2D and 3D generation, has received great attention. The goal of the NVS task is to generate multi-view images from reference images, and the core challenge is to maintain the 3D consistency between different view images. Recent works construct large 3D consistency multi-view image datasets and utilize the supervised fine-tuning (SFT) method to improve the 3D consistency. However, the SFT method suffers from the distribution shift, data inefficient problems, and lacks theoretical insight. To solve these problems, we discuss how to provide a suitable direction to the multi-view models and achieve better performance. More specifically, we first analyze the training-free guidance-based method and prove that contrastive guidance, which contains ground-truth and generated samples, can provide the right direction to improve 3D consistency. Based on the theoretical insight, we further design a contrastive 3D consistency metric and use it as the feedback in the following phase. To avoid the distribution shift problem, we use direct preference optimization (DPO) to fine-tune the multi-view diffusion models. Through qualitative and quantitative experiments, we demonstrate that after the fine-tuning phase with the above method, the 3D consistency of the multi-view images is significantly improved and achieves better performance compared to the SFT method.

## 1 Introduction

Recently, diffusion models have achieved impressive performance in 2D generation areas (Rombach et al., 2022; Podell et al., 2023; Karras et al., 2022) and show potential in the 3D generation domain (Poole et al., 2022; Zhang et al., 2024; Sun et al., 2023; Zhang et al., 2023). Since the 3D data is expensive and existing 3D datasets are not sufficient to directly train 3D diffusion models, many recent works focus on generating multi-view images of an object given a reference image, which is called novel view synthesis (NVS) (Liu et al., 2023b;a; Shi et al., 2023a; Long et al., 2024). We note that this task bridges 2D generation and 3D reconstruction. For the training phase, multi-view models can fully use the 2D prior information of large-scale diffusion models. For the 3D reconstruction task, a large reconstruction model uses the multi-view images as an input and outputs a 3D object.

An important point of the NVS task is to maintain the 3D consistency between different view images. There are two common methods to generate samples with certain properties, such as 3D consistency: (1) adding guidance to a pre-trained model and (2) fine-tuning the pre-trained model using a designed metric. Currently, most of existing multi-view diffusion models adopt the second method to improve the 3D consistency. More specifically, most works utilize supervised fine-tuning (SFT) with perfect 3D consistency multi-view image rendering from a 3D object (Liu et al., 2023a; Shi et al., 2023a; Liu et al., 2023b; Shi et al., 2023b; Long et al., 2024). Though the SFT method can improve the 3D consistency, it suffers from a distribution shift problem, which diminishes diversity and realism of the results (Li et al., 2023). Similar to the large language model, Xie et al. (2024) use reinforcement learning fine-tuning (RLFT) to improve the 3D consistency and avoid the distribution shift problem. However, their reward model relies heavily on NeRF reconstruction, which is time inefficient. To deal with this problem, Xie et al. (2024) use a closed-source sparse-view large reconstruction model (Li et al., 2023) to generate the 3D object, which is unfriendly to users. Furthermore, their 3D consistency

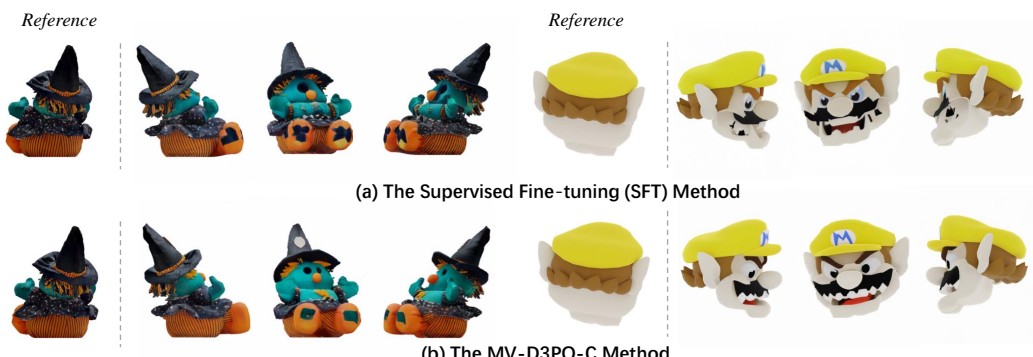

Figure 1: The qualitative results of our DPO method with contrastive 3D consistency feedback. Compared to the SFT method, the MV-D3PO-C method generates higher quality multi-view images.

metric leaks theoretical guarantee, and it is unclear which component plays an important role in 3D consistency. Hence, the following natural question remains open:

*It is possible to design a lightweight, effective, and theoretically interpretable 3D consistency feedback and use it to improve the 3D consistency of the pre-trained models?*

In this work, we start with the guidance-based method to provide the theoretical insight of 3D consistency. Then, we extend to the direct preference optimization (DPO)-based method and fine-tune the pre-trained model to achieve great performance. The guidance-based method aims to provide a correct direction to the pre-trained model without training. Though this method performs well in 2D generation areas (Ho and Salimans, 2022; Dhariwal and Nichol, 2021; Bansal et al., 2023), the suitable guidance for multi-view models to generate 3D consistent samples is unclear. In this work, for the first time, we study the the design principles of guidance in the 3D generation. We first show that the universal guidance Bansal et al. (2023), which only uses the self-generated samples, can not provide a correct direction for the 3D consistency. To solve this problem, we design a contrastive guidance and prove that it can improve the 3D consistency of multi-view diffusion models. More specifically, we view the ground-truth sample as the positive 3D consistency sample and self-generated samples as the negative samples. Then, we show that the positive samples increase the 3D consistency component, and the negative samples reduce the influence of other components, which leads a correct direction for improving the 3D consistency.

Though the guidance-based method is training-free, it can not improve the diffusion models unlimitedly (Guo et al., 2024). To achieve better performance, we design a 3D consistency feedback and use it to fine-tune the multi-view diffusion models. Since the reference image and the ground-truth images of other views render from the same object, these images are more consistent than the generated pair (reference and generated images). Inspired by our contrastive guidance, we view the ground-truth pairs as positive and generated pairs as negative pairs and train a contrastive feedback model. Since it is hard to guarantee absolute accuracy for trained feedback, we use the DPO method, which only requires a relative preference between different generated samples, to fine-tune the multi-view diffusion models. After the fine-tuning phase with the MV-D3PO-C method (the DPO method with the contrastive feedback for multi-view models), the 3D consistency of multi-view models has been significantly improved from qualitative and quantitative perspectives. In conclusion:

- We study the design principle of the suitable guidance for 3D consistency and prove that contrastive guidance with suitable positive and negative pairs can provide a right direction for improving 3D consistency.
- Inspired by the theoretical analysis, we design a contrastive metric to represent the 3D consistency. Taking advantage of the above metric, we provide the MV-D3PO-C method and show that it significantly improves the 3D consistency of multi-view models.

## 2 RELATED WORK

**Multi-View Diffusion Models.** Recently, many works focus on the NVS tasks and discuss how to improve the 3D consistency of multi-view images (Liu et al., 2023a; Shi et al., 2023a; Liu

et al., 2023b; Shi et al., 2023b; Long et al., 2024). Since they adopt the simple diffusion loss in the training phase, these works use the SFT method, and the main novelty of these works is to design the network architecture to share similar semantic content and texture with the reference images. More specifically, Zero123++ uses the reference-only attention mechanism to share reference image features. SyncDreamer proposes a 3D-aware feature attention mechanism (Liu et al., 2023b), MVdream propose an inflated 3D self-attention mechanism (Shi et al., 2023b), and Wonder3D uses a cross-domain attention mechanism (Long et al., 2024). However, the SFT method suffers from the distribution shift, and we need to provide a more generalizable method to improve the 3D consistency.

**RLFT and DPO for Diffusion Models.** Since the RLHF paradigm has achieve great success in the large language model (Ouyang et al., 2022; Rafailov et al., 2024), there is a series work using RLFT and DPO methods to fine-tune 2D diffusion models to improve some certain properties such as text2image alignment, body problem, human aesthetic (Xu et al., 2024; Black et al., 2023; Fan et al., 2024; Yang et al., 2024; Wallace et al., 2024). However, these works relies heavily on large dataset to train a reward model or use it as the preference feedback, which is expensive for the 3D generation. For multi-view diffusion models, as far as we know, only Xie et al. (2024) use the RLFT method to improve the 3D consistency of multi-view images. As discussed in Section 1, their 3D consistency reward relies heavily on the closed-source large reconstruction model and lacks theoretical insight.

**The Theoretical Guarantee for Conditional Diffusion Models.** Though the conditional diffusion models achieve great performance in application (Dhariwal and Nichol, 2021; Ho and Salimans, 2022; Bansal et al., 2023), there are a few works to analyze it from the theoretical perspective and these works focus on the guidance-based method (Wu et al., 2024; Guo et al., 2024). Wu et al. (2024) analyze the Gaussian mixture models and prove that the current guidance for diffusion models not only boosts classification confidence but also diminishes distribution diversity. Guo et al. (2024) analyze the guidance-based method from the optimization perspective and show that suitable guidance (Bansal et al., 2023) can preserve the structure of data distribution. Though these works deepen the understanding of the guidance-based method for 2D generation, they do not involve how to improve the 3D consistency, and we prove that the guidance proposed by Guo et al. (2024) can not provide a right direction for 3D consistency.

## 3 PRELIMINARIES

In this section, we first introduce the basic and notation of diffusion models from a continuous stochastic process perspective (Song et al., 2020). Then, we discuss how to add suitable guidance to the pre-trained diffusion models to generate samples with desired properties. Finally, we discuss how to extend the current 2D diffusion models to multi-view diffusion models

### 3.1 DIFFUSION MODELS

Let $q_0$ be the target data distribution. A diffusion model aims to generate samples from $q_0$ by an iterative denoising process consisting of forward and reverse processes. The forward process gradually converts the data distribution to pure Gaussian, and a common variance-preserving forward process is defined by:

$$\mathrm{d}X_t = -\frac{\beta(t)X_t}{2}\mathrm{d}t + \sqrt{\beta(t)}\mathrm{d}B_t \,, X_0 \sim q_0 \in \mathbb{R}^d \,,$$

where $\{B_t\}_{t\in[0,T]}$ is a $d$-dimensional Brownian motion and $\beta(t)$ is a non-dereasing non-negative function. After determining the forward process, the conditional distribution $X_t|X_0$ is a Gaussian $q_t(X_t|X_0) = \mathcal{N}(\alpha(t)X_0, h(t)I_d)$ with $\alpha(t) = \exp\left(-\int_0^t \beta(s)/2ds\right)$ and $h(t) = 1 - \alpha^2(t)$, which indicates $q_T$ is close to $\mathcal{N}(0, I_d)$ with a large forward diffusion time $T$.

To generate samples from pure noise, diffusion models reverse the forward process and obtain the reverse process:

$$\mathrm{d}X_t^{\Leftarrow} = [\beta(T-t)X_t^{\Leftarrow}/2 + \beta(T-t)\nabla \log q_{T-t}(X_t^{\Leftarrow})]\,\mathrm{d}t + \sqrt{\beta(T-t)}\,\mathrm{d}B_t \,, X_0^{\Leftarrow} \sim q_T \,,$$

where $(X_t^{\Leftarrow})_{t\in[0,T]} = (X_{T-t})_{t\in[0,T]}$. As shown in Cattiaux et al. (2021), the above process has the same density function $q_t$ with the corresponding forward process. Since $\nabla \log q_t(\cdot)$ contains the data

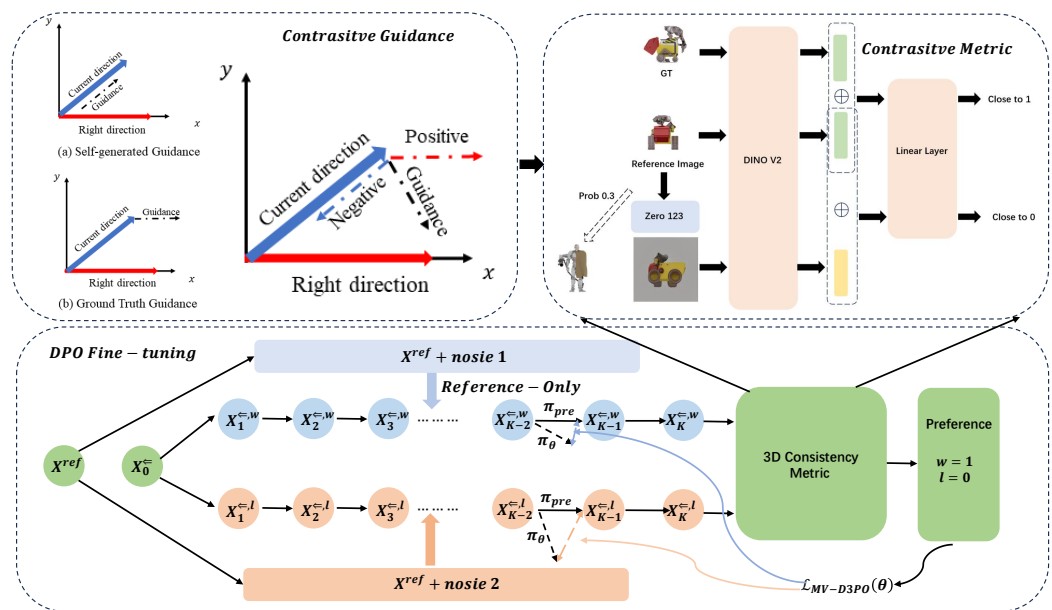

Figure 2: The pipeline of our MV-D3PO-C algorithm. **Top (Left)**: The theoretical intuition for the necessity of contrastive feedback for 3D consistency. **Top (Right)**: The training paradigm of contrastive 3D consistency metric, which views the ground-truth and reference images as the 3D consistent positive pairs and the generated and reference images as negative pairs. **Bottom**: The DPO fine-tuning phase with the contrastive feedback. Here, We use $k$ instead of $t_k^{\Leftarrow}$ for simplicity.

distribution information, diffusion models use the following score matching objective function (Song et al., 2020) to approximate it by using a neural network $s_\theta(\cdot, t)$:

$$\mathcal{L}_{\text{simple}} = \int_0^T \mathbb{E}_{X_0} \mathbb{E}_{X_t|X_0} \left[ \|\nabla_{X_t} \log q_t(X_t \mid X_0) - s_\theta(X_t, t)\|^2 \right] \mathrm{d}t.$$

Since a diffusion model can not run a continuous process, the model needs to discretize the process after obtaining the approximated score function. Let $t_0^{\Leftarrow} = 0 \leq t_1^{\Leftarrow} \leq ... \leq t_K^{\Leftarrow} = T - \delta$ be the discretization points in the reverse process. Then, we can run the following process to sample data:

$$\mathrm{d}\widehat{X}_t^{\Leftarrow} = \left[ \frac{\beta(T-t)\widehat{X}_t^{\Leftarrow}}{2} + \beta(T-t)s_\theta(\widehat{X}_{t_k^{\Leftarrow}}^{\Leftarrow}, t_k^{\Leftarrow}) \right] \mathrm{d}t + \sqrt{\beta(T-t)}\mathrm{d}B_t, \, t \in [t_k^{\Leftarrow}, t_{k+1}^{\Leftarrow}], \quad (1)$$

where $\widehat{X}_0^{\Leftarrow} \sim \mathcal{N}(0, I_d)$.

**The Guidance Diffusion Models.** To generate samples with a desired property such as 3D consistency, we can add a suitable guidance $G(\widehat{X}_{t_k^{\Leftarrow}}^{\Leftarrow}, t_k^{\Leftarrow}, g)$ to the reverse process (Guo et al., 2024):

$$\mathrm{d}\widehat{X}_t^{\Leftarrow} = \left[ \frac{\beta(T-t)\widehat{X}_t^{\Leftarrow}}{2} + \beta(T-t)(s_\theta(\widehat{X}_{t_k^{\Leftarrow}}^{\Leftarrow}, t_k^{\Leftarrow}) + G(\widehat{X}_{t_k^{\Leftarrow}}^{\Leftarrow}, t_k^{\Leftarrow}, g)) \right] \mathrm{d}t + \sqrt{\beta(T-t)}\mathrm{d}B_t,$$

where the guidance $G$ is determined by current $X$ and a direction (gradient) $g$ corresponds to the desired property. This method has achieved great performance in 2D controllable generation (Song et al., 2020; Ho and Salimans, 2022; Bansal et al., 2023) and is easy to implement. However, since it is a training-free method, it can not improve the diffusion model unlimitedly (Guo et al., 2024). Hence, we introduce the fine-tuning method for multi-view models in the next section.

## 3.2 THE FINE-TUNING BASED METHOD FOR MULTI-VIEW DIFFUSION MODELS

The section introduces two common fine-tuning methods: supervised fine-tuning (SFT) and direct preference optimization (DPO) methods. We adopt the SFT method, which uses the ground-truth

multi-view images as the supervised information, to obtain a base model in the pre-training phase. In the fine-tuning phase, we make full use of our designed contrastive 3D consistency feedback and DPO algorithm to improve 3D consistency.

### 3.2.1 THE SFT METHOD FOR MULTI-VIEW MODELS

The multi-view image generation task aims to generate 3D consistent $M$ different view images $\{X_0^{(1)}, X_0^{(2)}, ..., X_0^{(M)}\}$ given predefined camera pose and a reference image $Y \in \mathbb{R}^d$ (Fig. 1). For the goal of a multi-view diffusion model, we want to learn and generate a joint distribution $q(X_0^{(1:M)}|Y) := q(X_0^{(1)}, ..., x_0^{(M)}|Y)$. The most popular implementation is to adopt the SFT method by viewing the multi-view images rendered from an object as the target. More specifically, these works simply add noise to every view independently in the forward process, and the objective function for multi-view diffusion models becomes

$$\mathcal{L}_{\text{simple}-\text{MV}} = \int_0^T \mathbb{E}_{(X_0^{(1:M)}, Y)} \mathbb{E}_{Y_t|Y} \mathbb{E}_{X_t|X_0} \left[ \left\| \nabla \log q_t \left( X_t^{(1:M)} | X_0^{(1:M)} \right) - s_\theta \left( X_t^{(1:M)}, Y_t, t \right) \right\|^2 \right] \mathrm{d}t .$$

As discussed in Section 2, previous works focus on the network architecture design to share similar semantic content and texture with the reference images. Since the core of this work is not the design of the network architecture, we adopt the widely used reference attention mechanism proposed by Shi et al. (2023a). More specifically, the reference attention mechanism runs the U-net model on the noised reference image $Y_t$ and appends self-attention key and value matrices from the reference image to the corresponding attention layers from $X_t^{(1:M)}$. Similar to previous works, we also utilize an attention mechanism to propagate information across different views to generate consistent multi-view images. After obtaining $s_\theta(\cdot, \cdot, t)$, we can generate $M$ different view by the iterative denoising process simultaneously.

### 3.2.2 THE DPO METHOD FOR MULTI-VIEW MODELS

As shown in Xie et al. (2024), the SFT-based method suffers from the distribution shift problem, and they proposed a RLFT-based method to improve the 3D consistency of multi-view diffusion models. However, their reward model relies heavily on the large reconstruction models, which is unfriendly to users. In this work, we use the DPO-based method, which only requires relatively accurate feedback and is more direct and cost-effective, to fine-tune multi-view models. The first step for RLFT and DPO-based methods is to model the sampling phase of diffusion models as a markov decision process (MDP). Since our multi-view diffusion model is an extension of 2D diffusion models, we adopt the multi-step MDP modelling of Yang et al. (2024) [1]:

$$s_k \triangleq \left( c, k, X_{t_k^{\Leftarrow}}^{(1:M)} \right) \quad P\left(s_{k+1}|s_k, a_k\right) \triangleq \left( \delta_c, \delta_{t+1}, \delta_{X_{t_{k+1}^{\Leftarrow}}^{(1:M)}} \right)$$

$$a_k \triangleq X_{t_{k+1}^{\Leftarrow}}^{(1:M)} \qquad \pi_\theta\left(a_k|s_k\right) \triangleq q_\theta\left( X_{t_{k+1}^{\Leftarrow}}^{(1:M)}|c, t, X_{t_k^{\Leftarrow}}^{(1:M)} \right),$$

$$\rho_0\left(s_0\right) \triangleq \left(p(c), \delta_0, \mathcal{N}(0, I)\right)$$

$$r\left(s_k, a_k\right) \triangleq r\left( \left(c, t, X_{t_k^{\Leftarrow}}^{(1:M)}\right), X_{t_{k+1}^{\Leftarrow}}^{(1:M)} \right)$$

where $\delta_X$ represents the Dirac delta distribution, $k \in [1, K]$ is the denoising steps, $c$ is the conditional information and $q_\theta$ means run Eq. (1) with a score funciton $s_\theta$. As shown in Section 3.2.1, we view the attention parameters of $Y_t$ as the conditional information. After defining the above MDP, diffusion models with different $\theta$ can be viewed as different policies.

**DPO for Multi-view Diffusion Models.** After defining the above MDP, we can view diffusion models with different $\theta$ as different policies and use the DPO algorithm to optimize the pre-trained models. Let the pre-trained multi-view diffusion model be $\pi_{\text{pre}}$. As shown in Fig. 2, given a reference image $Y$ and the initial state $s_0^w = s_0^l = s_0$, we use the pre-trained multi-view diffusion model $\pi_{\text{pre}}$ generates two sequences $\sigma_w = \{s_0^w, a_0^w, ..., s_{K-1}^w, a_{K-1}^w, s_K^w\}$ and $\sigma_l = \{s_0^l, a_0^l, ..., s_{K-1}^l, a_{K-1}^l, s_K^l\}$, where $\sigma_w$ denotes the preferred sequence (the blue sequence in Fig. 2) and $\sigma_l$ denotes the inferior one

---

[1] In this part, we ignore the superscript $\Leftarrow$ and $\widehat{\cdot}$ of $X$ for simplicity.

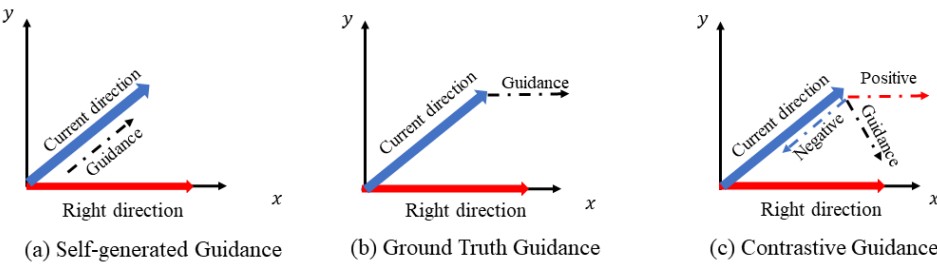

(a) Self-generated Guidance     (b) Ground Truth Guidance     (c) Contrastive Guidance

Figure 3: The explanation of our contrastive 3D consistency guidance

(the red sequence in Fig. 2). The preference between $\sigma_w$ and $\sigma_l$ is determined by our contrastive 3D consistency feedback (Top (Right) of Fig. 2 and Section 5). Since it is hard to distinguish the preference of the semi-finished multi-view images, similar to Yang et al. (2024), we obtain the preference by using the generated clean multi-view images $X_{0,w}^{(1:M)}$ and $X_{0,l}^{(1:M)}$. More specifically, we give reward by $r(s_k, a_k) = 1, \forall k \in [K]$ for winning the game and $r(s_k, a_k) = -1, \forall k \in [K]$ for losing the game. Then, we define the MV-D3PO (Multi-view Direct Preference Denoising Diffusion Policy Optimization) loss, which is extended from the D3PO loss proposed by Yang et al. (2024):

$$\mathcal{L}_{\mathrm{MV-D3PO}}(\theta) = -\mathbb{E}_{(s_i, \sigma_w, \sigma_l)} \left[ \log \rho \left( \beta \log \frac{\pi_\theta \left(a_i^w \mid s_i^w\right)}{\pi_{\mathrm{pre}} \left(a_i^w \mid s_i^w\right)} - \beta \log \frac{\pi_\theta \left(a_i^l \mid s_i^l\right)}{\pi_{\mathrm{pre}} \left(a_i^l \mid s_i^l\right)} \right) \right], \quad (2)$$

where the first component increases the likelihood of the preferred sequence and the second component decreases the likelihood of the dispreferred sequence. After that, the overall objective in the fine-tuning phase is defined as

$$\mathcal{L} = \mathcal{L}_{\mathrm{MV-D3PO}} + \mathcal{L}_{\mathrm{simple-MV}},$$

where $\mathcal{L}_{\mathrm{MV-D3PO}}$ is used to guide the pre-trained model to generate more 3D consistency samples and $\mathcal{L}_{\mathrm{simple-MV}}$ is a regularization term, which is used to prevent model degradation.

The remaining things are to design a 3D consistency metric and provide a correct direction for the optimization process, as discussed in the following two sections.

## 4 THE CONTRASTIVE GUIDANCE LEADS TO A CORRECT DIRECTION FOR THE 3D CONSISTENCY: A THEORETICAL PERSPECTIVE

In this section, we start from a 3D object generation task $X_0^{\mathrm{3D}} \sim q_0^{\mathrm{3D}} \in \mathbb{R}^D$ (In this work, we use $X^{\mathrm{3D}}$ and dimension $D$ represent 3D data (3D objects).) and discuss how to design suitable guidance for leading the 3D model to generate the required image given a certain view. This operation is similar to "rendering" a certain view image from a 3D object. If designed guidance can achieve this goal, it can generate 3D consistency multi-view images since these images come from the same object. Hence, we view the design of this guidance as the first step to improve 3D consistency.

Let $A = [a_1, a_2, ..., a_M] \in \mathbb{R}^{D \times M}$ be a column orthonormal matrix and $X_0^{\mathrm{3D}} \sim q_0^{\mathrm{3D}} \in \mathbb{R}^D$ be a 3D objective. Since $X^{\mathrm{3D}}$ can be viewed as a combination of $M$ different view images, we assume the following low-dimensional subspace assumption.

**Assumption 1.** *3D data can be represented as $X^{\mathrm{3D}} = Az$, where $A \in \mathbb{R}^{D \times M}$ is a column orthonormal matrix and $z \sim \mathcal{N}(\mu, \Sigma)$ is a Gaussian variable.*

The above assumption means different view images share a low-dimensional latent representation $z \in \mathbb{R}^M$, and each view corresponds to an eigenvector $a_i \in \mathbb{R}^D$. As shown in Chen et al. (2023), when the data distribution admits a linear subspace, the score function admits the following form

$$\nabla \log q_t(X^{\mathrm{3D}}) = A \nabla \log q_t^{\mathrm{LD}} \left(A^\top X^{\mathrm{3D}}\right) - \frac{1}{\sigma_t^2} \left(I_D - A A^\top\right) X^{\mathrm{3D}}, \quad (3)$$

where $q_t^{\mathrm{LD}} \left(Z'\right) = \int q_t \left(Z' | Z\right) q_z(Z) \mathrm{d}Z$ and $q_t(\cdot | Z) = \mathcal{N}(m_t Z, \sigma_t^2 I_M)$.

After assuming **Assumption 1**, the goal of the guidance phase is to lead the 3D diffusion model to generate images with the corresponding eigenvector when given a required view. We use the inner

product $a_i X^{\mathrm{3D}}$ to measure the similarity between $i$-th view and datapoint $X^{\mathrm{3D}}$. Let $\mathbb{E}\left[X_0^{\mathrm{3D}} \mid X_t^{\mathrm{3D}}\right]$ denotes the posterior expectation of clean data $X_0^{\mathrm{3D}}$ given $X_t^{\mathrm{3D}}$. Let $y^{\mathrm{Target}}$ be the upper bound of the similarity between $\{a_i\}_{i \in [1,M]}$ and $X_0^{\mathrm{3D}}$. Then, a trivial guidance to increase the similarity between the $i$-th view is proposed by Guo et al. (2024):

$$G_{\mathrm{Simple},i}\left(X_t^{\mathrm{3D}}, t, \tilde{a}_i\right) := -\nabla_{X_t^{\mathrm{3D}}}\left(y^{\mathrm{Target}} - \tilde{a}_i^\top \mathbb{E}\left[X_0^{\mathrm{3D}} \mid X_t^{\mathrm{3D}}\right]\right)^2,$$

where $\tilde{a}_i \in \{a_i, a_i^{\mathrm{appro}}\}$ corresponds to the clean data and the data generated by diffusion models, respectively. More specifically, we can do eigenvalue decomposition on the sample generated by the diffusion model to obtain it. However, since the 3D consistency of objects generated by pre-trained diffusion models needs to be improved, $a_i^{\mathrm{appro}}$ is not equal to $a_i$, and we want to make the generated samples more similar to $a_i$ given a certain view $i$. For the $\mathbb{E}\left[X_0^{\mathrm{3D}} \mid X_t^{\mathrm{3D}}\right]$, we note that given a score function, we obtain the following equality by using the Tweedie's formula and Eq. (3):

$$\mathbb{E}\left[X_0^{\mathrm{3D}} \mid X_t^{\mathrm{3D}}\right] = \frac{1}{\alpha(t)}\left(X_t^{\mathrm{3D}} + h(t)\nabla \log q_t\left(X_t^{\mathrm{3D}}\right)\right) = \frac{h(t)}{\alpha(t)} A m\left(A^\top X_t^{\mathrm{3D}}\right),$$

where $m(u)$ is a short notation of $\nabla \log q_t^{\mathrm{LD}}(u) + \frac{1}{h(t)}u$. In Corollary 1, we show that it is hard to use $G_{\mathrm{Simple},i}$ to guide 3D diffusion models to generate samples similar to $a_i$.

**Corollary 1.** *[Trivial guidance does not work.] Assuming* **Assumption 1***, we know that*

$$G_{\mathrm{Simple},i}\left(X_t^{\mathrm{3D}}, t, \tilde{a}_i\right) = \frac{2h(t)}{\alpha(t)}(y^{\mathrm{Target}} - \tilde{a}_i^\top \mathbb{E}\left[X_0^{\mathrm{3D}} \mid X_t^{\mathrm{3D}}\right])A\left[\nabla m\left(A^\top X_t^{\mathrm{3D}}\right)\right]^\top A^\top \tilde{a}_i,$$

*where $\tilde{a}_i \in \{a_i, a_i^{\mathrm{appro}}\}$.*

When assuming a Gaussian latent, as shown in Lemma 1 of Guo et al. (2024), the coefficient of $\tilde{a}_i$ is a positive-definite matrix. Hence, the above corollary indicates that the guidance direction is approximately the direction of $\tilde{a}_i$, which means using simple guidance would not generate samples similar to $a_i$. As shown in Fig. 3, when using the approximated eigenvector $a_i^{\mathrm{approx}}$ (Fig. 3 (a)) or true eigenvector $a_i$ (Fig. 3 (b)), we can not provide a correct direction. We also use Example 1 to make a clearer discussion.

**Example 1.** *We use a simple example $M = 2, D = 3$ to make a clearer discussion. In this case, we focus on $a_1$ and assume $a_1^\top = [1, 0, 0]$, $a_2^\top = [0, 1, 0]$. Due to the error introduced by generated models, we assume $a_1^{\mathrm{appro},\top} = [4/5, \sqrt{5}/5, 2/5]$, where the dominated term is still the first component. When considering $\tilde{a}_1 = a_1^{\mathrm{appro}}$, it is clear that though $G_{\mathrm{Simple},1}$ increase the first component, which corresponds to $a_1$, it increase the second and third components at the same time. When considering $\tilde{a}_1 = a_1$, the trivial guidance increase the first component and maintain the other components. Hence, this guidance can not lead models to generate samples similar with $a_1$.*

Based on the above observation, we design contrastive guidance, which can lead the model to generate samples similar to $a_i$ after fixing the $i$-th view:

$$G_{\mathrm{3D-Consistency},i}\left(X_0^{\mathrm{3D}}, X_t^{\mathrm{3D}}, t\right) := -\nabla_{X_0^{\mathrm{3D}}}\left(y^{\mathrm{Target}} - a_i^\top X_0^{\mathrm{3D}}\right)^2 - \nabla_{X_t^{\mathrm{3D}}}\left(a_i^{\mathrm{appro},\top} \mathbb{E}\left[X_0^{\mathrm{3D}} \mid X_t^{\mathrm{3D}}\right]\right)^2.$$

Since the ground-truth samples are 3D consistent, we view them as a positive sample. For the samples generated by the diffusion models, 3D consistency is weaker than ground-truth samples, and we view them as negative pairs. Theorem 1 shows that this guidance can provide a correct direction to generate samples similar to $a_i$ given view $i$.

**Theorem 1.** *[Contrastive Guidance provide a correct direction.] Assuming* **Assumption 1***, we have*

$$G_{\mathrm{3D-Consistency},i}\left(X_0^{\mathrm{3D}}, X_t^{\mathrm{3D}}, t, a_i, a_i^{\mathrm{appro}}\right)$$

$$= 2(y^{\mathrm{Target}} - a_i^\top X_0^{\mathrm{3D}})a_i - \frac{2h(t)}{\alpha(t)} A\left[\nabla m\left(A^\top X_t^{\mathrm{3D}}\right)\right]^\top A^\top a_i^{\mathrm{appro}}.$$

As shown in Theorem 1, $G_{\mathrm{3D-Consistency}}$ is consisted of two parts: $a_i$ and $-a_i^{\mathrm{appro}}$. The first part provides the correct direction for the $i$-th component and reduces the influence of other components. Let $\alpha_{1,i}$ be the positive coefficient of $a_i$ and $\alpha_{2,i}$ be the positive coefficient of $a_i^{\mathrm{appro}}$. Note that we can guarantee the $\alpha_{1,i} = \Theta(\alpha_{2,i})$ by carefully choosing $y^{\mathrm{Target}}$ or adding additional $\frac{h(t)}{\alpha(t)}$. Then, $G_{\mathrm{3D-Consistency}}$ becomes $\alpha_{1,i}a_i - \alpha_{2,i}a_i^{\mathrm{appro}}$. We still use Example 1 and Fig. 3 (c) to make a clearer discussion. The direction of guidance is $[\alpha_{1,1} - \alpha_{2,1}4/5, -\alpha_{2,1}\sqrt{5}/5, -\alpha_{2,1}2/5]$, which increase the first component and decrease the other components. Hence, our designed guidance provides a correct direction for the 3D diffusion model.

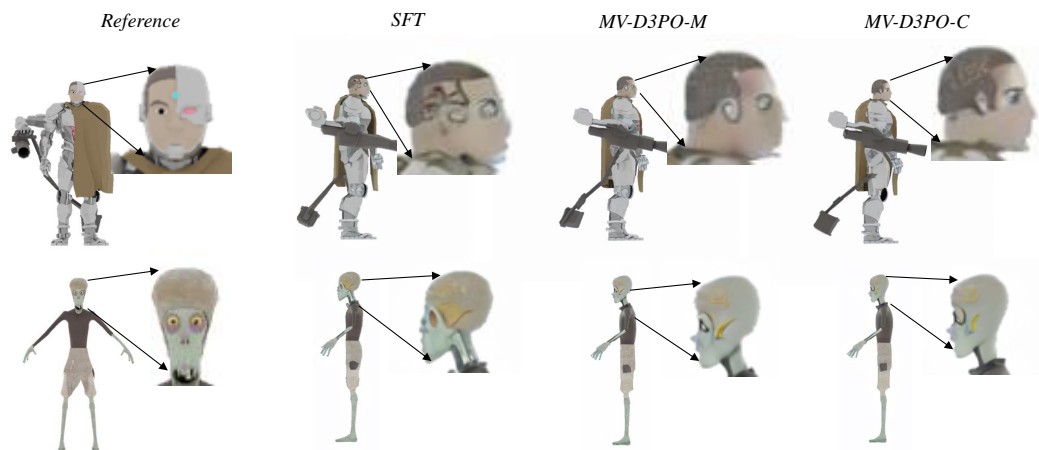

Figure 4: Typical 3D consistency eyes problem for the SFT method.

## 5 CONTRASTIVE 3D CONSISTENCY FEEDBACK FOR MULTI-VIEW MODELS

As shown in the above section, contrastive 3D guidance can help a 3D object to render a certain view. Since the different view images rendering from a 3D object is 3D consistent, we think the intuition of contrastive guidance can be shared with the multi-view generation task. However, the multi-view generation task does not have a generated 3D object, which means we can not calculate $\nabla_{X_0^{3D}}$ and $\nabla_{X_t^{3D}}$. An executable solution sharing a similar idea is to design a 3D consistency feedback and fine-tune the pre-trained multi-view models. In this part, we show the procedure to obtain a contrastive 3D consistency feedback.

The contrastive training paradigm requires positive and negative pairs. To guarantee the 3D consistency of positive pairs, we use ground-truth multi-view images and the reference image as positive pairs. Then, we use images generated by Zero123 (Liu et al., 2023a) and reference images as the negative samples. Since DINOv2 has shown its 3D awareness (El Banani et al., 2024), we use it as an encoder to obtain the embedding of the reference images $e_Y$, the ground-truth other view images $e_{X_{\mathrm{GT}}^m}$ and other view images generated by zero123 $e_{X_0^m}$. Then, we use a linear layer $f$ to calculate the similarity between the reference and other view images and minimize the following loss function to obtain our evaluation model:

$$\mathcal{L}_{3\mathrm{D-Consistency}} = \|f(e_Y, e_{X_{\mathrm{GT}}^m}) - \mathbf{1}\|_2^2 + \|f(e_Y, e_{X_0^m}) - \mathbf{0}\|_2^2.$$

In the training phase of the contrastive metric, we fine-tune DINOv2 and the linear layer to achieve a better result. In the evaluation phase, we calculate the contrastive score between the reference image and $m$-th generated other view image and sum up as our final contrastive score.

We show that the contrastive score trained on the Zero123 and ground-truth images can effectively distinguish between ground-truth other view images and images generated by other multi-view diffusion models. As shown in Table 1, the contrastive score for the ground-truth images is 2.14, which is higher than the contrastive score of all other multi-view images. Furthermore, the contrastive score of our baseline models, which is better on the classic metrics, is also higher than SyncDreamer.

## 6 EXPERIMENTS

In this section, we show that the MV-D3PO-C method and its modified versions have better performance and significantly improve the 3D consistency compared with the SFT method from the qualitative and quantitative perspective.

### 6.1 EXPERIMENT PROTOCOL

**The Pre-trained Multi-view Models.** For the multi-view data, we render multi-view images of an object from Objaverse (Deitke et al., 2023). Since the orthographic view is enough for reconstructing

Table 1: The quantitative results in novel view synthesis task. We report LPIPS, SSIM, PSNR and our contrastive score.

| Method | LPIPS ↓ | SSIM ↑ | PSNR ↑ | CL score ↑ |
|---|---|---|---|---|
| SFT | 0.053 | 0.9321 | 27.3582 | 2.1071 |
| MV-D3PO-M | **0.0517** | 0.9342 | 27.5475 | 2.0983 |
| MV-D3PO-C (pre) | 0.0521 | 0.9336 | 27.48 | 2.1078 |
| MV-D3PO-C | **0.0517** | **0.9346** | **27.5969** | **2.1155** |
| SyncDreamer | 0.1648 | 0.8434 | 17.3952 | 2.0748 |
| Ground-truth | - | - | - | 2.1481 |

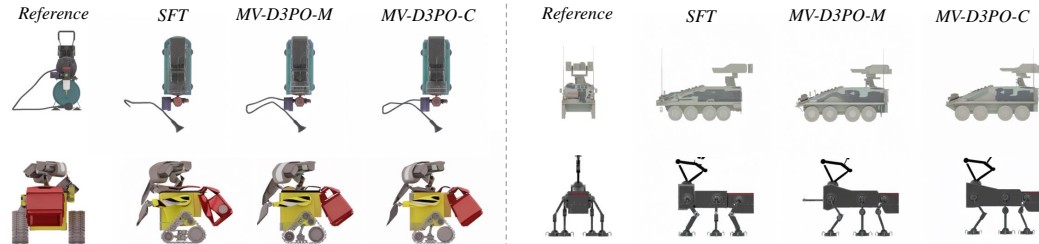

Figure 5: Typical 3D consistency multi-object problem for the SFT method.

a 3D object, we choose five orthographic views: front, right, back, left and top. In this work, we only use one image as the reference, which matches the requirement of the empirical application. For the network architecture, we add the reference attention mechanism proposed by Shi et al. (2023a) to the Stable Diffusion XL (Podell et al., 2023). Finally, we randomly choose an image from the front, right, back and left view images as the reference image and train the multi-view diffusion model by using $\mathcal{L}_{\mathrm{simple-MV}}$.

**Baseline.** To do a fair comparison, we also fine-tune the pre-trained model using the SFT method under the same optimization step and training dataset. To show the role of contrastive 3D consistency feedback, we propose two modified versions of MV-D3PO: MV-D3PO-M and MV-DPO-C (pre).

Recall that SyncDreamer Liu et al. (2023b) runs COLMAP (Schönberger et al., 2016) on the generated multi-view images and reports the reconstructed 3D point number as the 3D consistency feedback, which requires more than 10 images. However, we only generate the orthographic view images, which is not enough to run this algorithm. In this work, we also propose a matching-based 3D consistency feedback, which is suitable for sparse multi-view images. More specifically, we calculate the matching points between the reference image and other view images by using the existing matching method Sun et al. (2021). Then, we view generated samples with more matching points as better 3D consistency. When using this matching feedback, the algorithm is MV-D3PO-M.

For the MV-D3PO-C (pre) method, we train a contrastive metric with the pre-trained models and use it as the feedback to fine-tune the pre-trained model. We also provide the quantitative results of SyncDreamer (Liu et al., 2023b). However, due to the different rendering parameters, their results can not be directly compared with ours.

**Metrics.** For the novel view synthesis, we first adopt three commonly used metrics, i.e. LPIPS (Zhang et al., 2018), SSIM (Wang et al., 2004) and PSNR. To further evaluate the 3D consistency of multi-view images, we also use our contrastive 3D consistency metric (Section 5, Zero123 version) to evaluate the generated images. The evaluation dataset consists of 200 objects randomly sampled from the Objaverse dataset. The experiment detail and more qualitative results are shown in Appendix B.

## 6.2 DISCUSSION.

This part discusses the experiment results from the qualitative and quantitative perspectives. From a qualitative standpoint, we focus on two 3D consistency problems of the SFT method: the eye problem (Fig. 4) and the multi-object problem (Fig. 5). The eye problem means that it is difficult for the model

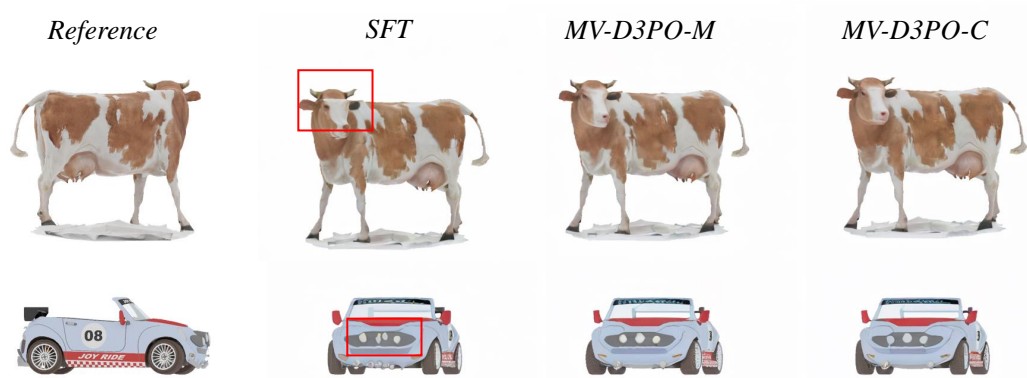

Figure 6: Typical 3D consistency problem for the SFT method.

to generate realistic pupils or glasses. The multi-view problem means the model prefers to generate broken objects or objects with holes given a reference object.

**The Role of Direct Preference Optimization.** As shown in Table 1, the DPO-based methods improve the classic metric compared to the SFT method. Since our training dataset only contains 240 datapoints, this result indicates that the DPO-based method (including the matching and contrastive feedback) is more data efficient and has generalizability. In other words, the DPO-based method can alleviate the distribution shift problem of the SFT method. The qualitative results also show that the MV-D3PO-based methods generated high-quality multi-view images compared to the SFT method (Fig. 4 and Fig. 5).

**The Contrastive 3D Consistency Feedback.** As shown in Table 1, the contrastive score of MV-D3PO-M is lower, and the score of MV-D3PO-C and C (pre) is higher than the SFT method. This observation further supports our theoretical analysis and indicates that the feedback of the DPO-based method needs to be carefully designed to improve the classic metric and our contrastive 3D consistency score simultaneously. The empirical observation also supports the above discussion. More specifically, Fig. 4 shows that though the face of MV-D3PO-M is more realistic than the results of the SFT method, the generated eyes are worse than the one of MV-D3PO-C.

## 7 CONCLUSION

In this work, we study the design principle of 3D consistency feedback of multi-view diffusion models from the theoretical and empirical perspectives at the same time. For the theoretical insight, we analyze the guidance-based method and design contrastive guidance consisting of ground-truth images and generated images. Then, we prove that it is necessary to use contrastive guidance instead of previous universal guidance to provide a correct direction for improving the 3D consistency.

To achieve better performance in application, we go beyond the training-free guidance-based method, propose the DPO-based method MV-D3PO and show that this method can alleviate the distribution shift problem of the SFT method. For the MV-D3PO method, we propose two modified versions, the matching feedback version and the contrastive feedback version and show that the multi-view images generated by the MV-D3PO-C are more 3D consistent compared to MV-D3PO-M, which also supports our theoretical results.

**Future work and Limitation.** In this work, we show the DPO-based method can effectively improve the 3D consistency of multi-view diffusion models with suitable feedback. Except for the 3D consistency problem, there are also some other problems, such as distorted fingers or facial features. It is an interesting future work to design suitable feedback for these properties and use the MV-D3PO method to solve these problems. Furthermore, our work focuses on the 3D consistency of the multi-view diffusion model. It is an interesting future work to achieve an end-to-end 3D consistency 3D reconstruction by using a well-trained large sparse-view large reconstruction model with our contrastive 3D consistency feedback.

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

APPENDIX

# A  THE PROOF OF THE CONSISTENCY 3D CONSISTENCY METRIC

**Corollary 1.** *[Trivial guidance does not work.] Assuming* **Assumption 1**, *we know that*

$$G_{\text{Simple},i}\left(X_t^{\text{3D}}, t, \tilde{a}_i\right) = \frac{2h(t)}{\alpha(t)}(y^{\text{Target}} - \tilde{a}_i^\top \mathbb{E}\left[X_0^{\text{3D}} \mid X_t^{\text{3D}}\right])A\left[\nabla m\left(A^\top X_t^{\text{3D}}\right)\right]^\top A^\top \tilde{a}_i,$$

*where* $\tilde{a}_i \in \{a_i, a_i^{\text{appro}}\}$.

**Proof.** We first expand the derivative in $G_{\text{Simple},i}$ and obtain the following equation:

$$G_{\text{Simple},i}\left(X_t^{\text{3D}}, t, \tilde{a}_i\right) = 2\left(y^{\text{Target}} - \tilde{a}_i^\top \mathbb{E}\left[X_0^{\text{3D}} \mid X_t^{\text{3D}}\right]\right)\left(\nabla_{X_t^{\text{3D}}} \mathbb{E}\left[X_0^{\text{3D}} \mid X_t^{\text{3D}}\right]\right)^\top \tilde{a}_i.$$

Theorem 1 of Guo et al. (2024) show that the following equation holds when assuming **Assumption 1**

$$\nabla_{X_t^{\text{3D}}} \mathbb{E}\left[X_0^{\text{3D}} \mid X_t^{\text{3D}}\right] = \frac{h(t)}{\alpha(t)}A\left[\nabla m\left(A^\top X_t^{\text{3D}}\right)\right]^\top A^\top, \tag{4}$$

where $m(u)$ is a short notation of $\nabla \log q_t^{\text{LD}}(u) + \frac{1}{h(t)}u$. Then, we complete our proof. ∎

**Theorem 1.** *[Contrastive Guidance provide a correct direction.] Assuming* **Assumption 1**, *we have*

$$G_{\text{3D}-\text{Consistency},i}\left(X_0^{\text{3D}}, X_t^{\text{3D}}, t, a_i, a_i^{\text{appro}}\right)$$

$$=2(y^{\text{Target}} - a_i^\top X_0^{\text{3D}})a_i - \frac{2h(t)}{\alpha(t)}A\left[\nabla m\left(A^\top X_t^{\text{3D}}\right)\right]^\top A^\top a_i^{\text{appro}}.$$

**Proof.** We also expand the derivative in $G_{\text{3D}-\text{Consistency},i}\left(X_0^{\text{3D}}, X_t^{\text{3D}}, t\right)$

$$G_{\text{3D}-\text{Consistency},i}\left(X_0^{\text{3D}}, X_t^{\text{3D}}, t\right) = 2(y^{\text{Target}} - a_i^\top X_0^{\text{3D}})a_i - 2\left(\nabla_{X_t^{\text{3D}}} \mathbb{E}\left[X_0^{\text{3D}} \mid X_t^{\text{3D}}\right]\right)^\top a_i^{\text{appro}}.$$

Then, we can use Eq. (4) and achieve the final results. ∎

# B  EXPERIMENT DETAIL

**The Construction of Different Multi-view Pairs.**  Since our multi-view model is not a text-to-3D model, we can not construct negative pairs by using negative text prompts (Yang et al., 2024). To generate different multi-view pairs with the same reference images, as shown in Fig. 2, we add different scale noise to the embedding of reference images. In this work, we add 5 scale pure noise $a_{\text{noise}}\mathcal{N}(0, I)$ with $a_{\text{noise}} \in \{0, 0.1, 0.3, 0.5, 0.8\}$ to the embedding of the reference image.

**Training Detail of MV-D3PO.**  For the pre-train model, we fine-tune all parameters of reference-only attention and Stable Diffusion XL on the Objaverse dataset for $400$ A100 days.

For the MV-D3PO fine-tuning phase, we first run the pre-trained model to collect data and construct different multi-view pairs. More specifically, we run the pre-trained models with different noise scale reference images and save $\pi_{\text{pre}}(a_k|s_k), \forall k \in [K]$ information. In this process, we collect 240 datapoints as the training dataset of the fine-tuning phase. After that, we fine-tune the attention layers of Stable Diffusion XL and reference-only attention for $1800$ optimization steps, which need 4 hours on 1 V100. Similarly to the ImageReward (Xu et al., 2024), we only fine-tune the last 40 denoising steps of the denoising step.

## B.1  MORE EXPERIMENTS RESULTS

In this part, we provide more results on the 3D consistency of the SFT method and show the great performance of our MV-D3PO-C method.

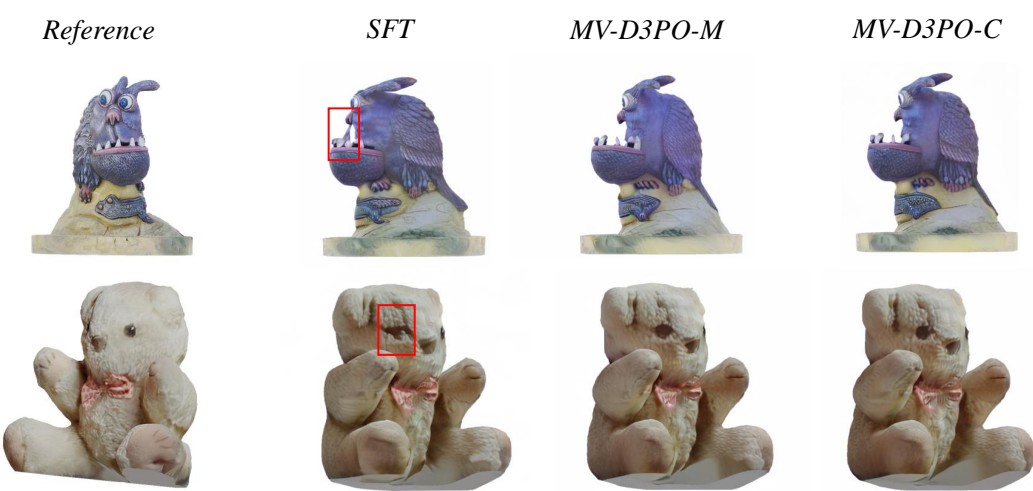

Figure 7: Typical 3D consistency problem for the SFT method (More examples).

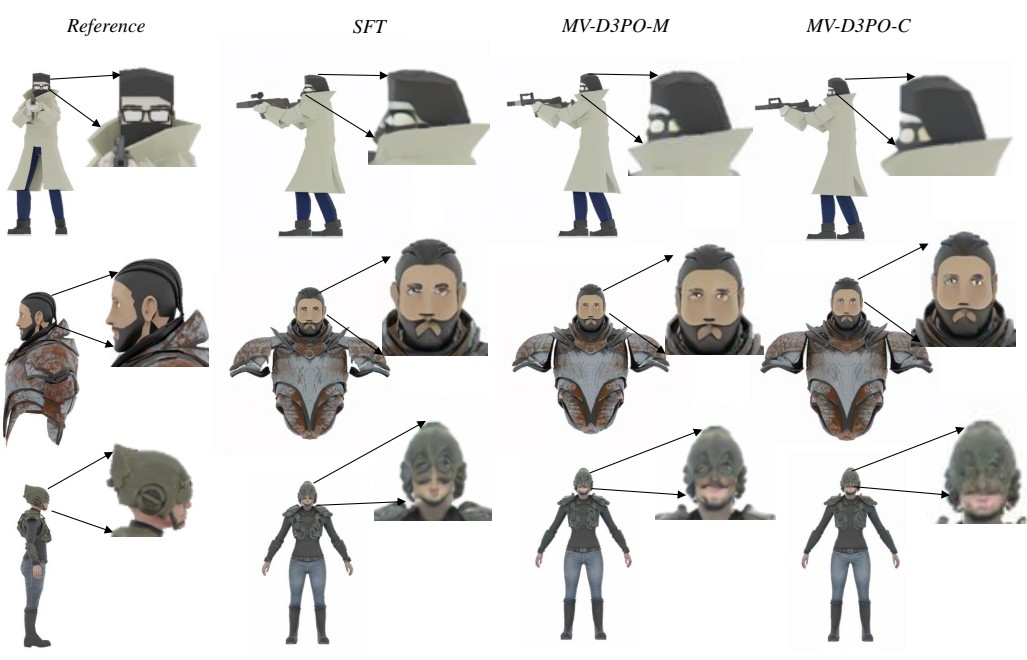

Figure 8: Typical 3D consistency eyes problem for the SFT method (More examples).

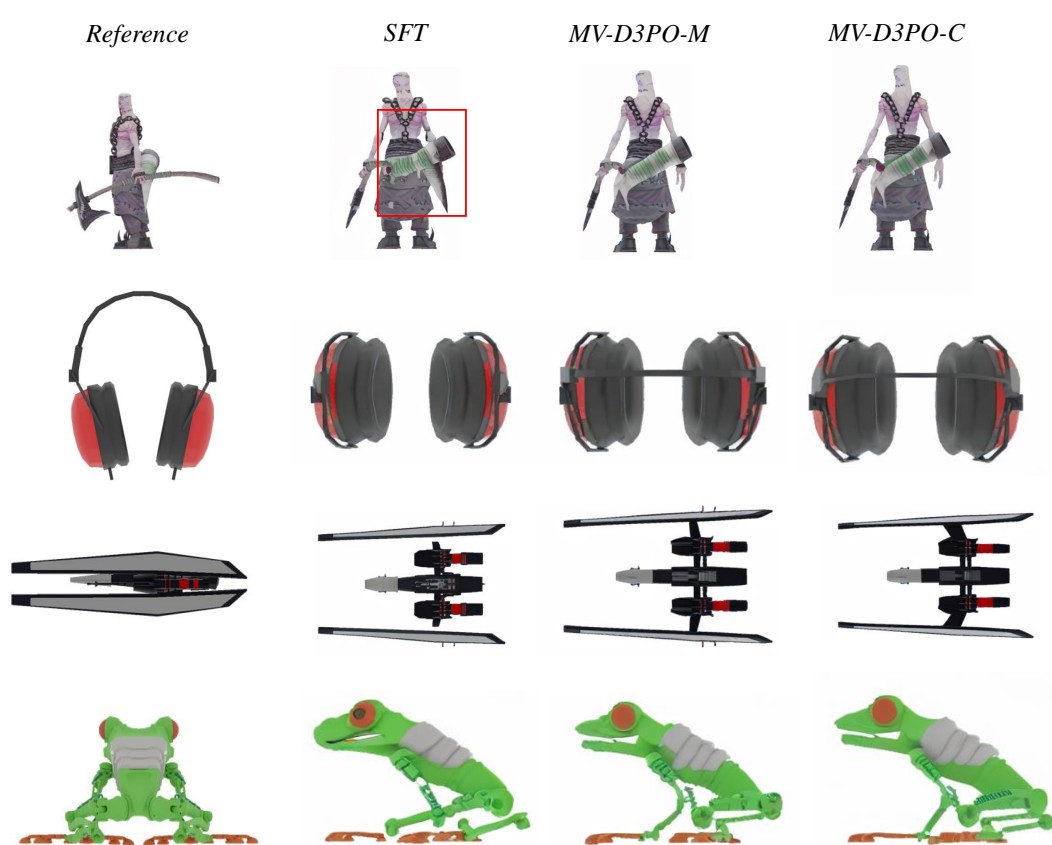

Figure 9: Typical 3D consistency problem (More examples).

