# OpenReview forum: "Contrastive guidance and feedback: A Suitable way to improve 3D Consistency of Multi-view Diffusion Model"
_ICLR.cc/2025/Conference — Submitted to ICLR 2025_

### Official Review · Reviewer_UYPQ · 2024-10-29

**Soundness:** 2
**Presentation:** 1
**Contribution:** 2
**Rating:** 3
**Confidence:** 3

**Summary:**

This paper proposes a new method to improve the multiview consistency of multiview diffusion models.
The proposed method first trains a multiview diffusion model using 3D ground-truth data.
Then, the method is further optimized with a direct preference optimization (DPO) loss without requiring ground truth.
This DPO will utilize a contrastive learning-based model to determine the preference of the generated images, where multiview consistent images will be preferred.
Experiments demonstrate the DPO improves the consistency in the multiview generation.

**Strengths:**

The paper combines the multiview diffusion model (MVDream, SyncDreamer), DPO (D3PO), and contrastive learning to design a new multiview generation framework. The framework is novel to me.

**Weaknesses:**

1. The main weakness is the writing of the paper. I can understand Section 3, which introduces the existing techniques like diffusion, D3PO, and multiview diffusion. Section 4 seems to be the key idea of the paper but I cannot understand it.
1) What is X^{3D}? I'm not sure whether this means a 3D model or multiview images here.
2) Why X^3D can be represented by a linear transformation of Gaussian noise in assumption 1? Both definitions seem not to be valid here.
After that, I totally get lost in the rest discussion of the section.
2. Experiments are not convincing enough. The best way to evaluate the multiview consistency is to conduct 3D reconstruction from the generated multiview images and check the quality of the 3D reconstruction. However, only some NVS metrics like PSNR are provided in the experiments.
3. Meanwhile, the experiment lacks comparison with recent multiview diffusion models like Era3D, Zero123++, etc. Does the proposed framework also work on these new models?
4. The proposed method seems to be an incremental combination of three existing works. The motivation for introducing the RLFT in multiview diffusion models is to resolve the domain-shifting problem. However, the evaluation seems still to be synthetic images, which does not support the claim here.

**Questions:**

Refer to weaknesses.

---

> ### Author Response · Authors · 2024-11-25
> **Part 1**
>
> Thank you for your valuable comments and suggestions. We provide our response to each question below. **Please click the PDF to view our revision paper.**
>
> **Weakness 1 (a): The presentation.**
>
> Thanks for the helpful comments on our paper. We have polished our presentation in our revision paper  (highlighted in green). More specifically, we make a clearer explanation of the multi-view diffusion models and our MV-D3PO algorithm in Section 3. In Section 4 and 5, we provide the intuition of contrastive guidance (for 3D generation) and contrastive feedback (for multi-view image generation).
>
> **Weakness 1 (b)&2 (a). The discussion on the 3D contrastive guidance part.**
>
> The $X^{3D}$ means a 3D object with dimension $D$. For the 2D image, we use $d$ as its dimension (Hence, the $M$ different view images have dimension $M\times d$)  Since the 3D object can be viewed as a combination of multi-view images ($A$ in Assumption 1) and different view images share the same latent information ($z$ in Assumption) (such as 3D representation), we have Assumption 1. For the linear assumption, it is widely used in theoretical works [1] [2].
>
> In this part, we make a clearer discussion on the role of contrastive guidance and the relationship with our contrastive feedback. For the guidance section, we focus on the 3D generation task and discuss how to use suitable guidance to "render" a certain view from a 3D object (which is generated by a 3D diffusion model). In this process, we introduce the intuition of contrastive guidance.
>
> Since the different view images rendering from a 3D object is 3D consistent, we think the intuition of contrastive guidance can be shared with the multi-view generation task. However, the multi-view generation task does not have a generated 3D object, which means we can not calculate $\nabla\_{X\_0^{3D}}$ and $\nabla\_{X\_t^{3D}}$ (two core parts of the contrastive guidance). An executable solution sharing a similar idea is to design a contrastive 3D feedback (Sec.5) and fine-tune the pre-trained multi-view models  (Sec. 6). As shown in the experiment part, our MV-D3PO-C achieve a great performance from the qualitative and quantitative perspective. We have now added the above discussion at the beginning of Section 5 in our revision paper (highlighted in green).
>
> **Weakness 2 (b). The 3D consistency.**
>
> The generated quality of different view images is guaranteed by the classical NVS metrics such as LIPIPS, SSIM and PSNR. For the 3D consistency, as shown in our multi-object cases (Figure 5), the samples generated by the pre-trained models contain multi-object, which means the 3D object (reconstructed by a large reconstruction model given different view images) still has multiple parts. On the contrary, our generated samples only have one object, which is more 3D consistent compared with the pre-trained models. For the eyes case, the multi-view images generated by pre-trained models lose the feature of the pupil. On the contrary, our samples preserve the feature.
>
> Furthermore, our CL score (which gives a higher score for the 3D consistency pairs) also shows that our generated samples have a higher score.
>
> **Weakness 3. The additional experiments.**
>
> Thanks for the comments on the additional experiment. We also test the performance of zero123++, which achieves LIPIPS 0.318, SSIM 0.7812, PSNR 9.0268. Though the result of zero123++ is worse than the results in Table 1, we note that these results can not be directly compared with our results due to the different rendering parameters (as we discussed at the end of the baseline part). We note that our contrastive 3D feedback and DPO framework is uniformly applicable to multi-view diffusion models and it is an interesting future work to show universe improvement in different multi-view models.

---

> > ### Author Response · Authors · 2024-11-25
> > **Part 2**
> >
> > **Weakness 4: The role of RLFT in multi-view diffusion models.**
> >
> > The only existing work for multi-view diffusion models with RLFT is CARVE3D [3]. As shown in the abstract of CARVE3D, their goal is also to solve the multi-view inconsistency problem. In the following paragraph, we give a detailed comparison between CARVE3D and our MV-D3PO-C algorithm.
> >
> > The CARVE3D use a sparse-view large reconstruction model (for example, they use closed-source Instant3D) to determine their reward. Then, they use DDPO (a PPO-based algorithm) to fine-tune the pre-trained model.
> >
> > For the first point, their algorithm relies on a high-quality large reconstruction model (LRM), which requires large high-quality data and is unfriendly to users. Though it is possible to use open-source LRM (such as OpenLRM and LGM), the performance of DPPO with these models is unclear and the LRM models are large. On the contrary, our contrastive 3D feedback is lightweight and only uses $1000$ data to fine-tune the DINOV2 model.
> >
> > For the second point, our method inherits the advantages of DPO and is data-efficient (as we discuss in the response to Weakness 4).
> >
> > For the experiment results, since CARVE3D does not release their checkpoint, we do not provide their empirical performance with our test dataset.
> >
> > [1] Chen, M., Huang, K., Zhao, T., & Wang, M. (2023, July). Score approximation, estimation and distribution recovery of diffusion models on low-dimensional data. In International Conference on Machine Learning (pp. 4672-4712). PMLR.
> >
> > [2] Yuan, H., Huang, K., Ni, C., Chen, M., & Wang, M. (2024). Reward-directed conditional diffusion: Provable distribution estimation and reward improvement. Advances in Neural Information Processing Systems, 36.
> >
> > [3] Xie, D., Li, J., Tan, H., Sun, X., Shu, Z., Zhou, Y., ... & Kaufman, A. E. (2024). Carve3d: Improving multi-view reconstruction consistency for diffusion models with rl finetuning. In *Proceedings of the IEEE/CVF Conference on Computer Vision and Pattern Recognition* (pp. 6369-6379).

---

> > > ### Comment · Reviewer_UYPQ · 2024-11-26
> > >
> > > Thanks for your response.
> > >
> > > However, I'm still confused about why we can have an assumption that the multiview data can be generated by $Az$ where $A$ contains some bases and $z$ is a Gaussian noise. The whole theory is based on this assumption but I think the multiview images follow their own probability distributions like epipolar constraints. Epipolar constraints cannot be described by the assumption.
> > >
> > > The paper looks like a combination of several techniques, DPO, MVD, and contrastive learning. It seems that the main contribution is to utilize the above assumption to use contrastive learning in the pipeline here.
> > >
> > > The results are not impressive enough without any 3D reconstruction results and even a video.
> > >
> > > I'm still on the negative side and want to reject this paper.

---

> > > > ### Author Response · Authors · 2024-12-03
> > > >
> > > > We thank the reviewers for the detailed reply. The 3D reconstruction from the multi-view images is a great way to verify the 3D consistency of generated images. We will add the 3D reconstruction (qualitative and quantitative) results in our next version paper.

---

### Official Review · Reviewer_wbo9 · 2024-11-02

**Soundness:** 3
**Presentation:** 3
**Contribution:** 3
**Rating:** 6
**Confidence:** 3

**Summary:**

This paper presents a novel approach to enhance multi-view diffusion models using Direct Preference Optimization (DPO). The authors first analyze training-free guidance methods, demonstrating that contrastive guidance in real and generated images during the guidance process helps improve 3D consistency. By using ground-truth multi-view images and the reference image as positive pairs,  images generated by Zero123  and reference images as the negative samples, and utilizing a fine-tuned DINOv2 model as evaluation model. Both qualitative and quantitative experiments validate the effectiveness of their DPO-based method.

**Strengths:**

- The authors provide a fully theoretical proof to demonstrate the effectiveness of their method, and the experimental results also provide strong support for their theory.
-  Instead of using large reconstruction models as reward model, this paper just use the diffusion model's ouptuts as negative samples. It's quite simple and efficient, which can be easily adapted to other models.
- The visual results demonstrate that the method can effectively addresses a key limitation in multi-view generation, where models trained directly with SFT fail to generate reasonable views. This is a significant problem in the field.

**Weaknesses:**

-  Both methods (SFT and DPO) score about the same on technical metrics like LPIPS, SSIM, and PSNR. Adding some human feedback evaluation would help us get a better picture of which one actually works better.

**Questions:**

Would using different models (besides Zero123), such as Zero123++, as sources of negative samples produce different effects on the results? Since SyncDreamer's base model was trained on Zero123, would using outputs from other models have any significant impact?

---

> ### Author Response · Authors · 2024-11-25
>
> #
>
> ---
>
> Thank you for your valuable comments and suggestions. We provide our response to each question below.
>
> **Weakness 1. The 3D consistency.**
>
> Thanks again for the suggestion on the human feedback evaluation on the 3D consistency property, we will add this part in our next version paper. In this part, we show how to support the discussion "our MV-D3PO-C algorithm achieves better 3D consistency".
>
> The generated quality of different view images is guaranteed by the classical NVS metrics such as LIPIPS, SSIM and PSNR. For the 3D consistency, as shown in our multi-object cases (Figure 5), the samples generated by the pre-trained models contain multi-object, which means the 3D object (reconstructed by a large reconstruction model given different view images) still has multiple parts. On the contrary, our generated samples only have one object, which is more 3D consistent compared with the pre-trained models. For the eyes case, the multi-view images generated by pre-trained models lose the feature of the pupil. On the contrary, our samples preserve the feature. Furthermore, our CL score (which gives a higher score for the 3D consistency pairs) also shows that our generated samples have higher scores.
>
> **Q1: The source of negative samples.**
>
> We note that as long as contrastive feedback is used, the 3D consistency property would be improved (with different negative models). As shown in Table 1, MV-D3PO-C (pre) and MV-D3PO-C all achieve better 3D consistency (CL score), where the first algorithm uses our pre-trained multi-view models as the negative model and the second algorithm uses Zero123 as the negative model. However, we also note that the choice of the negative models is also important. As shown in Table, though MV-D3PO-C (pre) achieve a better CL score compared with MV-D3PO-M, the other metrics are worse. On the contrary, the MV-D3PO-C (with zero123 as the negative model) achieves better results for all metrics.

---

### Official Review · Reviewer_QCXu · 2024-11-03

**Soundness:** 3
**Presentation:** 3
**Contribution:** 3
**Rating:** 5
**Confidence:** 4

**Summary:**

This work addresses the challenge of maintaining 3D consistency in the novel view synthesis (NVS) task. The authors analyze a training-free, guidance-based method and demonstrate that contrastive guidance can effectively enhance 3D consistency. Based on this insight, they design a contrastive 3D consistency metric and incorporate it as feedback in the subsequent phase. Additionally, a Direct Preference Optimization (DPO) procedure is proposed to fine-tune multi-view diffusion models. Qualitative and quantitative results are presented, comparing the proposed method to the supervised fine-tuning (SFT) approach, demonstrating its effectiveness.

**Strengths:**

The motivation of the paper is clear and well-founded, with a detailed theoretical analysis provided. The solution derived by the authors is theoretically sound and logically consistent.

The paper is well-written and presented.

**Weaknesses:**

As 3D consistency is a primary focus of this work, the absence of reconstruction from synthesized multi-view images limits the evaluation of 3D consistency. This type of assessment is crucial to demonstrate the robustness of the proposed approach in maintaining consistent geometry across multiple views.

The showcased results are limited, focusing primarily on the eye and multi-object problems. To better evaluate the method's effectiveness, objects with more complex appearances and geometries should also be tested.

**Questions:**

The authors avoided running COLMAP on the generated multi-view images, stating that they only produced orthographic views, which were insufficient for the algorithm. However, why not generate a greater number and variety of images to ensure sufficient data for running COLMAP and fully evaluate the 3D consistency of the method?

The authors adopt DINOv2 to obtain embeddings of the reference image. Would using an image encoder with a stronger 3D prior knowledge (e.g., CroCo or DUSt3R) lead to improved performance?

---

> ### Author Response · Authors · 2024-11-25
>
> Thank you for your valuable comments and suggestions. We provide our response to each question below. **Please click the PDF to view our revision paper.**
>
> **Weakness 1. The 3D consistency.**
>
> The generated quality of different view images is guaranteed by the classical NVS metrics such as LIPIPS, SSIM and PSNR. For the 3D consistency, as shown in our multi-object cases (Figure 5), the samples generated by the pre-trained models contain multi-object, which means the 3D object (reconstructed by a large reconstruction model given different view images) still has multiple parts. On the contrary, our generated samples only have one object, which is more 3D consistency compared with the pre-trained models. For the eyes case, the multi-view images generated by pre-trained models lose the feature of the pupil. On the contrary, our samples preserve the feature.
>
> Furthermore, our CL score (which gives a higher score for the 3D consistency pairs) also shows that our generated samples have a higher score.
>
> It is an interesting future work to achieve an end-to-end 3D reconstruction by using our generated multi-view images and well-trained large sparse-view large reconstruction model with contrastive 3D consistency feedback  (with our rendering parameters). We have added this discussion in the future work part of our revision paper (highlighted in green).
>
> **Weakness 2: the eyes and multi-object problem.**
>
> As discussed in Weakness 1, these two problems are typical and easy-to-observe problems for the 3D consistency property. We have added more examples with more complex geometries in Appendix B.1 of our revision paper.
>
> **Q1: The discussion on the COLMAP.**
>
> Since we use the reference attention technique, we need to generate different view images at the same time. Since the resolution of our images is $512*512$, it is hard to generate more than $10$ different view images simultaneously in a single A100. Hence, we choose five typically orthographic views to show our improvement compared with the pure SFT method.
>
> **Q2: the image encoder with a stronger 3D prior.**
>
> Thanks for your helpful comments! The image encoder of our contrastive 3D feedback can be any encoder with suitable 3D prior and we use DINOV2 as an example in this work. It is an interesting future work to use an image encoder with stronger 3D prior to achieving more fine-grained feedback.

---

> > ### Comment · Reviewer_QCXu · 2024-12-03
> >
> > Thanks for the response and the revision. After reading the rebuttal and other reviewers' opinions, I want to keep my original rating. My major concern still lies in the 3D consistency and I agree with reviewers UYPQ that 3D reconstruction and video results would help qualitatively evaluate 3D consistency, which are currently absent.

---

> > > ### Author Response · Authors · 2024-12-03
> > >
> > > We thank the reviewers for the detailed reply and comments. The 3D reconstruction from the multi-view images is a great way to verify the 3D consistency of generated images. We will add the 3D reconstruction (qualitative and quantitative) results and the results of current SOTA multi-view models in our next version paper.

---

### Official Review · Reviewer_TJrM · 2024-11-03

**Soundness:** 2
**Presentation:** 2
**Contribution:** 2
**Rating:** 5
**Confidence:** 3

**Summary:**

The paper proposes a novel approach to enhance 3D consistency in multi-view diffusion models by applying Direct Preference Optimization (DPO) with a specially designed 3D consistency metric that employs a contrastive strategy. Additionally, it provides theoretical proof for the use of contrastive guidance, demonstrating that trivial guidance cannot guarantee 3D consistency.

**Strengths:**

1. The application of DPO-based optimization and designing 3D consistency metric to multi-view diffusion models is both novel and intriguing.

2. A strong theoretical foundation for the use of a contrastive strategy, which is particularly valuable given that 3D generation often lacks comprehensive theoretical grounding.

**Weaknesses:**

1. Experimental validation of 3D guidance:
i) Showing the extent of performance improvement and the limitations of using only guidance would support the argument that while 3D guidance achieves better consistency than trivial guidance, it is not sufficient, necessitating fine-tuning.
It would be helpful to include experimental results demonstrating how well 3D guidance works during the inference phase.

ii) Additionally, if 3D guidance is effective, why does it not work well on pre-trained multi-view diffusion models without additional fine-tuning? It would be interesting to see how this guidance performs when applied to a general pretrained multi-view diffusion model. Clarifying the general applicability of guidance would add value.

2. Clarity of the connection between Guidance and Fine-tuning:
While the role of guidance is well-explained, a clearer explanation of how it connects to the use of contrastive loss during fine-tuning would be beneficial. Enhancing the logical connection between guidance and fine-tuning could help readers better understand how these two components synergize. As mentioned in line 397, transitioning from stating that guidance cannot achieve ultimate improvements to using fine-tuning may feel abrupt and could be explained more smoothly.

3. Complexity in writing:
The writing in the paper is somewhat difficult to follow. It might be helpful to explain the DPO for the Multi-view Diffusion Models section earlier in conjunction with Figure 2 to improve clarity.

4. The advantages of the proposed method are not clear:
i) The experimental setup does not fully demonstrate the advantages of the proposed method. Specifically, it would be valuable to explain clearly how using DPO + contrastive loss outperforms existing multi-view diffusion methods in terms of performance or resource efficiency.

ii) It is also unclear why the experiments do not include comparisons with the latest SOTA models and instead only use models trained by the authors. There are several models that outperform SyncDreamer, and comparing against these would provide a stronger case for the proposed method. Additionally, a comparison with CARV3D, which uses RLFT, would be essential for a more comprehensive evaluation.

**Questions:**

1. Could you clarify what is meant by "unfriendly to users" in 053?
2. In line 689, it is mentioned that the training took "400 A100 days."
3. The images in Figure 2 appear to include examples such as construction vehicles and forklifts. It is correct to add two different object types in Figure 2?

---

> ### Author Response · Authors · 2024-11-25
>
> Thank you for your valuable comments and suggestions. We provide our response to each question below. **Please click the PDF to view our revision paper.**
>
> **Weakness 1&2. The discussion on the 3D contrastive guidance.**
>
> Thanks for the comments on the contrastive 3D guidance and feedback. In this part, we make a clearer discussion of the role of these terms. For the guidance section, we focus on the 3D generation task and discuss how to use suitable guidance to "render" a certain view from a 3D object (which is generated by a 3D diffusion model). In this process, we introduce the intuition of contrastive guidance.
>
> Since the different view images rendering from a 3D object is 3D consistent, we think the intuition of contrastive guidance can be shared with the multi-view generation task. However, the multi-view generation task does not have a generated 3D object, which means we can not calculate $\nabla\_{X\_0^{3D}}$ and $\nabla\_{X\_t^{3D}}$ (two core parts of the contrastive guidance). An executable solution sharing a similar idea is to design a contrastive 3D feedback (Sec.5) and fine-tune the pre-trained multi-view models  (Sec. 6). As shown in the experiment part, our MV-D3PO-C achieve a great performance from the qualitative and quantitative perspective. We have now added the above discussion at the beginning of Section 5 in our revision paper (highlighted in green).
>
> **Weakness 3: The presentation.**
>
> Thanks again for the helpful comments on the introduction of DPO for multi-view diffusion models. We have polished our presentation of this part (please see Sec. 3.2 of our revision paper, highlighted in green).
>
> **Weakness 4: the advantage of the proposed MV-D3PO-C algorithm.**
>
> For the performance, we show that our algorithm achieves greater quantitative results (LIPIPS, SSIM, PSNR and CL score). We also show that detailed cases (the eyes and multi-object problem) show great performance from the qualitative perspective.
>
> For the resource efficiency perspective, as shown in Sec. 6.2, since the DPO-based method is more data-efficient,  we only use $240$ datapoints to fine-tune the pre-trained models, which can run in one V100. On the contrary, CARVE3D (a PPO-based method) uses $48$ A100 to fine-tune the pre-trained method
>
> **Weakness 5. The additional experiments.**
>
> Thanks for the comments on the additional experiment. We also test the performance of zero123++, which achieves LIPIPS 0.318, SSIM 0.7812, PSNR 9.0268. Though the result of zero123++ is worse than the results in Table 1, we note that these results can not be directly compared with our results due to the different rendering parameters (as we discussed at the end of the baseline part). We note that our contrastive 3D feedback and DPO framework is uniformly applicable to multi-view diffusion models and it is an interesting future work to show universe improvement in different multi-view models.
>
> **Weakness 5 \&Q1: The comparison with CARVE3D.**
>
> The CARVE3D use a sparse-view large reconstruction model (for example, they use closed-source Instant3D) to determine their reward. Then, they use DDPO (a PPO-based algorithm) to fine-tune the pre-trained model.
>
> For the first point, their algorithm relies on a high-quality large reconstruction model (LRM), which requires large high-quality data and is unfriendly to users. Though it is possible to use open-source LRM (such as OpenLRM and LGM), the performance of DPPO with these models is unclear and the LRM models are large. On the contrary, our contrastive 3D feedback is lightweight and only uses $1000$ data to fine-tune the DINOV2 model.
>
> For the second point, our method inherits the advantages of DPO and is data-efficient (as we discuss in the response to Weakness 4).
>
> For the experiment results, since CARVE3D does not release their checkpoint, we do not provide their empirical performance, such as PSNR, SSIM, etc.
>
>
>
> **Q2: The training time of the pre-trained models.**
>
> The 400 A100 days is the time to train our pre-trained models. For our MV-D3PO-C fine-tuning algorithm, as shown in the response of Weakness 4, our method is data and time-efficient.
>
> **Q3: The different objects**
>
> The forklift is the ground-truth other view image corresponds to the reference images. The "vehicles" is the other view image generated by zero123++ given the reference images. The "vehicles" image did not contain the forklift feature may be because Zero123  does not capture this feature.

---

> > ### Comment · Reviewer_TJrM · 2024-12-02
> >
> > I appreciate the effort required for the revision but I will maintain my previous assessment.
> >
> > The connection between contrastive guidance and fine-tuning could be better clarified to strengthen the methodological contributions. Additionally, as Reviewer UYPQ also pointed out, the manuscript still lacks sufficient experimental validation, such as 3D reconstruction results or detailed comparisons with recent SOTA models. I share the concern that these omissions limit the work's impact.

---

> > > ### Author Response · Authors · 2024-12-03
> > >
> > > We thank the reviewers for the detailed reply. The 3D reconstruction from the multi-view images is a great way to verify the 3D consistency of generated images. We will add the 3D reconstruction (qualitative and quantitative) results in our next version paper.

---

### Comment · Area_Chair_E9jZ · 2024-11-25
**Please read the rebuttal and reply**

Dear Reviewers,

Thanks again for serving for ICLR, the discussion period between authors and reviewers is approaching (November 27 at 11:59pm AoE), please read the rebuttal and ask questions if you have any. Your timely response is important and highly appreciated.

Thanks,

AC

---

### Meta-Review · Area_Chair_E9jZ · 2024-12-19

**Metareview:**

This paper proposes a method to enhance 3D consistency in multi-view synthesis diffusion models through contrastive guidance, it additionally provides theoretical analysis of the proposed method in addition to experimental evidence. The paper is novel and proposes an interesting solution. During rebuttal, the main concern raised by reviewers is that the submission lacks sufficient results to support its claim. Specifically, to verify 3D consistency of multi-view images, one important way is to use these images as supervision in 3D reconstruction, yet the submission has not provided that. Other issues raised by reviewers include complex writing, missing comparison with SOTA methods. After rebuttal, 3 out of 4 reviewers remain their initial score while 1 reviewer did not respond. Considering this, the paper is not ready for publication at this stage and is recommended for rejection.

**Additional Comments On Reviewer Discussion:**

During rebuttal, the reviewers raised the following concerns:
- the submission lacks sufficient results to support its claim. Specifically, to verify 3D consistency of multi-view images, one important way is to use these images as supervision in 3D reconstruction, yet the submission has not provided that.
- complex writing, there are multiple repeated words and typos throughout the paper
- missing comparison with SOTA methods.

After rebuttal, 3 out of 4 reviewers kept their initial negative score while 1 reviewer did not respond.

---

### Decision · Program_Chairs · 2025-01-22

Reject